# Ecological Stoichiometric Characteristics in Organs of *Ammopiptanthus mongolicus* in Different Habitats

**DOI:** 10.3390/plants12020414

**Published:** 2023-01-16

**Authors:** Xue Dong, Jinbo Zhang, Zhiming Xin, Yaru Huang, Chunxia Han, Yonghua Li, Qi Lu

**Affiliations:** 1Institute of Desertification Studies, Chinese Academy of Forestry, Beijing 100091, China; 2Experimental Center of Desert Forestry, Chinese Academy of Forestry, National Long-Term Scientific Research Base of Comprehensive Control in Ulan Buh Desert, Inner Mongolia Dengkou Desert Ecosystem National Observation Research Station, Dengkou, Bayannur 015200, China

**Keywords:** desert evergreen shrub, stoichiometric characteristics, organ difference, heterogeneous habitat, trade-off strategies, nutrient use strategy

## Abstract

The essence of plant ecological stoichiometry is to study the relationships between species and their environment, including nutrient absorption, utilization and cycling processes as well as the nutrient limitation of plants. Plants can regulate nutrient elements and adapt to environmental changes. To understand the adaptation mechanism, it is important to take plants as a whole and quantify the correlation between the chemometrics of different organs. *Ammopiptanthus mongolicus* is within the second-class group of rare–endangered plants in China and is the only evergreen broad-leaved shrub in desert areas. We analyzed the ecological stoichiometric characteristics of leaves, stems, roots, flowers and seeds of *A. mongolicus* in five habitats, namely fixed sandy land, semi-fixed sandy land, stony–sandy land, alluvial gravel slope and saline–alkali land. We found that (1) the nutrient contents of N, P and K were in the order of seed > flower > leaf > root > stem. The enrichment of the N, P and K in the reproductive organs promoted the transition from vegetative growth to reproductive growth. Additionally, (2) the contents of C, N, P and K and their stoichiometric ratios in different organs varied among different habitat types. The storage capacity of C, N and P was higher in sandy soil (fixed and semi-fixed sandy land), whereas the content of K was higher in gravelly soil (stony–sandy land and alluvial gravel slope), and the C:N, C:P and N:P were significantly higher in gravelly soil than those in sandy soil. *A. mongolicus* had higher nutrient use efficiency in stony–sandy land and alluvial gravel slope. Furthermore, (3) the C:N and N:P ratios in each organ were relatively stable among different habitats, whereas the K:P ratio varied greatly. The N:P ratios of leaves were all greater than 16 in different habitats, indicating that the growth was mainly limited by P. Moreover, (4) except for the P element, the content of each element and its stoichiometric ratio were affected by the interaction between organs and habitat. Habitat had a greater impact on C content, whereas organs had a greater influence on N, P and K content and C:N, C:P, C:K and N:P.

## 1. Introduction

Carbon (C), nitrogen (N), phosphorus (P) and potassium (K) are the basic chemical nutrients for plant growth and development, and they regulate important physiological and ecological processes such as plant photosynthetic rate, transpiration rate, reproduction and growth [1,2,3,4]. Ecological stoichiometric characteristics can reflect the homeostasis of plant organs and the distribution ratio and interrelationship of each element in different organs [5,6]. Stoichiometric ratio can also determine the limiting elements in the process of plant growth and nutrient use efficiency [7,8,9,10]. Through the study of the ecological stoichiometric characteristics of plant organs, the nutrient recycling status in the process of plant growth, reproduction, regeneration and restoration can be grasped. Previous studies have shown that there is a certain correlation between the ecological stoichiometric characteristics of plant leaves, stems and roots at the individual level [11,12]. The stable metabolism and growth and development of plants reflect the response and adaptation of plants to environmental changes [13,14]. Therefore, the study of the ecological stoichiometric characteristics of plant carbon, nitrogen, phosphorus and potassium is helpful to understand the growth regulation mechanism and survival strategy of plants, and to explore the effects of plant organs and heterogeneous habitats on plants.

Under global climate change, drought in northwest China has intensified and desertification has expanded. Ecological adaptation of plants to desertification conditions has become a key scientific issue. *Ammopiptanthus mongolicus* (Maxim. ex Kom.) Cheng f, as a tertiary relic species, is among the first batch of rare and endangered plants under national key protection. The species is mainly distributed in central and western Inner Mongolia, northern Ningxia and northern Gansu, and the most concentrated and suitable areas are the desert margins and piedmont alluvial gravelly slopes in the Western Ordos–East Alxa region [15,16]. *A. mongolicus* is a leguminous species and the only evergreen broad-leaved shrub in the arid desert areas of China. It has a special classification status and is extremely resistant to drought and heat; thus, it plays an important role as a windbreak and in sand fixation in harsh and barren desert areas. Different organs of plants have their specific functions, growth and turnover rates, resulting in differences in nutrient content among organs [8]. The complex feedback relationship between organisms and climate and terrain can regulate the characteristics of C:N:P:K [17]. Site type is a complex of site factors such as geology, climate, hydrology and soil microenvironment. The same site conditions have similar resources and environment, so similar tending methods and nutrient diagnosis methods can be used for operation and management. Different sites are often affected by differences in light, heat, rainfall, temperature and soil nutrients to form specific habitat, which can change the external resource supply and the availability of nutrient elements. Therefore, it further affects the adaptive strategies of plants [13], which in turn leads to the change of C, N, P and K involved in the nutrient cycle [14,18]. Studying the distribution pattern of stoichiometric characteristics of the same plant species in different habitat types is of important theoretical and practical significance for practitioners to select suitable forest sites, cultivate seedlings and manage forest nutrients.

Numerous studies have focused on plant chemometric characteristics from the perspectives of organs, species, communities and ecosystems. The influence of climate, topography, latitude and other factors on the chemometric characteristics of ecosystems has been discussed. However, research on ecological stoichiometry of plant organs focuses mainly on the content of carbon, nitrogen and phosphorus in leaves and their stoichiometric ratios, although research on the ecological stoichiometric characteristics of roots is increasing. In contrast, little is known about the stoichiometric characteristics of reproductive organs. As the components of plants, each organ should be regarded, but the distribution ratio of chemical elements among different plant organs and the correlation of stoichiometric characteristics between organs are still lacking. Therefore, this study analyzed the C, N, P, and K element contents and their stoichiometric ratio characteristics of the roots, stems, leaves, flowers and seeds of *A. mongolicus* under different habitat conditions and determined the nutrient distribution and nutrient distribution of plants from the perspective of plant organ ecological stoichiometry. This can help understand the response and adaptation mechanism of *A. mongolicus* to the environment and provide a scientific basis for explaining why it became endangered.

## 2. Results

### 2.1. Changes of C, N, P and K Content of Each Organ of A. mongolicus

There were certain differences in the contents of C, N, P and K and their ratios in each organ of *A. mongolicus* under different habitat conditions (Figure 1). The content of C in different organs showed as follows: leaf > stem > root > seed > flower, and the C content in leaves was significantly higher than that in stems (*p* < 0.05). The C content in stems was significantly higher than that in stems (*p* < 0.05), roots, seeds and flowers (*p* < 0.05), but the C content of roots, seeds and flowers was not significantly different (*p* > 0.05) (Figure 1A). The contents of N, P and K under different habitat conditions were as follows: seeds > flower > leaf > root > stem, in which the contents of N and K elements in different organs were significantly different (*p* < 0.05). There were no significant differences between seeds and flowers, roots and stems (*p* > 0.05). However, seeds and flowers were significantly higher than leaves (*p* < 0.05), whereas roots and stems were significantly lower than leaves (*p* < 0.05) (Figure 1B–D).

The contents of C, N, P and K in organs were different among different habitats. Among them, C in leaves and stems and P in leaves, flowers and seeds showed the following order: fixed sandy land > saline–alkali land > semi-fixed sandy land > alluvial gravel slope > stony–sandy land. The N in leaves and stems was as follows: saline–alkali land > fixed sandy land > semi-fixed sandy land > alluvial gravel slope > stony–sandy land (Figure 1A,C). N in seeds and flowers was expressed as: fixed sandy land > semi-fixed sandy land > alluvial gravel slope > stony sand land > saline–alkali land (Figure 1B). K in leaves, stems, roots and seeds showed the following order: stony–sandy land > alluvial gravel slope > fixed sandy land > semi-fixed sandy land > saline–alkali land (Figure 1D). The variation of C content in plant organs among different habitats was relatively small, whereas the variation coefficients of N, P and K were relatively large. The distribution ratio of nutrient elements in each organ of *A. mongolicus* was significantly different under different habitat conditions. Among them, the content of C, N, P and K in the underground part (root) accounted for 12.09%, 13.85%, 3.46% and 42.07% of the element content of the whole plant, respectively. The content of C, N, P and K in the aboveground parts (leaves and stems) accounted for 39.76%, 31.92%, 76.48% and 46.84% of the element content of the whole plant, respectively, indicating that the element content in the aboveground part was higher than that in the underground part.

The contents of C, N, P and K and their stoichiometric ratios in different organs of *A. mongolicus* under different habitats changed more complex and had no general pattern. C in shrubs showed the following order: fixed sandy land > semi-fixed sandy land > saline–alkali land > alluvial gravel slope > stony–sandy land (Figure 1A). N in shrubs showed the following order: fixed sandy land > semi-fixed sandy land > alluvial gravel slope > saline–alkali land > stony–sandy land (Figure 1B). P in shrubs showed the following order: fixed sandy land > saline–alkali land > semi-fixed sandy land > stony–sandy land > alluvial gravel slope (Figure 1C). K in shrubs showed the following order: stony–sandy land > alluvial gravel slope > fixed sandy land> semi-fixed sandy land > saline–alkali land (Figure 1D).

### 2.2. Characteristics of Chemical Element Stoichiometry in Each Organ of A. mongolicus

The descriptive statistics of the numerical values of C:N, C:P, C:K, N:P, N:K and K:P of each organ of *A. mongolicus* in different habitats is shown in Figure 2. The ratio of C:N was the lowest in seeds (10.26 ± 1.41) and the highest in stems (27.49 ± 3.25), respectively. There was no significant difference between leaves and roots (*p* > 0.05), but both of them were significantly higher than flowers (*p* < 0.05). The trend of variation in C:P and C:K in each organ was the same as that of C:N, and the ratio was as follows: stem > root > leaf > flower > seed (Figure 2A–C). The ratio of N:P was 22.82 ± 3.14 in roots, 21.84 ± 6.09 in leaves and 19.61 ± 5.21 in stems, among which the difference was not significant (*p* > 0.05), and the ratio of N:P was 9.04 ± 1.74 in flowers and 9.04 ± 1.74 in seeds. The median ratio was 13.11 ± 3.03 (Figure 2D). The ratio of N:K was larger in stem and root organs with no significant difference (*p* > 0.05) and was centered in leaves and significantly greater than that in seeds and flowers (*p* < 0.05) (Figure 2E). The variation range of K:P in each organ was 3.16–6.69. Specifically, the ratio in stem and flower organs was small, and there was no significant difference (*p* > 0.05) for the largest in leaf. The ratio was also relatively similar in roots and seeds (*p* > 0.05) (Figure 2F).

The stoichiometric ratios of C, N, P and K in organs were different among different habitats, and the C:N and K:P in leaves and stems were all in the following order: stony–sandy land > alluvial gravel slope > semi-fixed sandy land > fixed sandy land > saline–alkali land (Figure 2A,F). C:P and N:P in leaves were in the following order: alluvial gravel slope > stony–sandy land > semi-fixed sandy land > saline–alkali land > fixed sandy land (Figure 2B,D). C:K and N:K in leaves and roots were in the following order: saline–alkali land > semi-fixed sandy land > fixed sandy land > alluvial gravel slope > stony–sandy land (Figure 2C,E). The ratio of C:N and N:P in each organ was relatively stable among habitats, but K:P varied greatly among the habitats. C:N, C:P and K:P in shrubs showed the following order: stony–sandy land > alluvial gravel slope > semi-fixed sandy land > saline–alkali land > fixed sandy land (Figure 2A,B,F). C:K and N:K in shrubs showed the following order: saline–alkali land > fixed sandy land > semi-fixed sandy land > alluvial gravel slope > stony–sandy land (Figure 2C,E). N:P in shrubs showed the following order: alluvial gravel slope > semi-fixed sandy land > stony–sandy land > fixed sandy land > saline–alkali land (Figure 2D).

### 2.3. Effects of Organs and Habitats on the Elemental Content and Stoichiometric Ratios of A. mongolicus

The contents and stoichiometric ratios of C, N, P and K elements in *A. mongolicus* were affected by the interaction of single and double factors of organs and habitats to different degrees (Table 1). The effect of organs and habitats on C content reached a significant level, and the effect of habitats on C content was greater. The effects of organs on C:N, C:P and N:P all reached extremely significant levels, whereas the effects of habitats were not significant. The effects of organs and habitats on N, P and K content and C:K, N:K and K:P all reached extremely significant levels. K and K:P were mainly affected by the habitat. The interaction of habitats and organs had a significant effect on the contents of C, N and K elements, as well as C:N, C:K, N:K, N:P, K:P and C:P ratios. There was no significant effect on P.

## 3. Discussion

### 3.1. Variation Characteristics of C, N, P and K Contents and Stoichiometric Ratios in Each Organ of A. mongolicus

The average contents of C and N in leaves of *A. mongolicus* were 522.46 g/kg and 27.95 g/kg, which were higher than that of global terrestrial plants (464.00 g/kg, 20.60 g/kg) [19], eastern China North–South transect (480.10 g/kg, 18.30 g/kg) [20], China grassland ecosystem (438.00 g/kg, 27.60 g/kg) [21], Alxa (379.01 g/kg, 10.65 g/kg) [22], Loess Plateau (438.30 g/kg, 24.10 g/kg) [23] and East Alxa–West Ordos (435.07 g/kg, 23.40 g/kg) [24]. However, the average contents of P and K in leaves of *A. mongolicus* were 1.41 g/kg and 7.32 g/kg, which were lower than the average levels of P (1.50 g/kg) and K (15.09 g/kg) in leaves of Chinese terrestrial plants [25]. Compared with other study areas, the contents of C and N in *A. mongolicus* were higher, whereas the contents of P and K were lower. Usually, the higher the C content in the leaves, the lower the photosynthetic rate, the slower the growth rate, and the stronger the defense ability against the harsh environment [26]. As an evergreen broad-leaved shrub species in desert areas, its growth was relatively slow; it produced enough organic matter for itself through photosynthesis. The growth of plants in arid desert areas was vulnerable to water stress. Increasing the N content of leaves makes it possible for plants to adapt to the arid environment [27]. Under drought stress, the increase of proline and other substances could increase the osmotic pressure of cells, thereby mitigating the impact of drought [28], so the N content increased. Nitrogen fixation by nodules played an important role in the growth and development of *A. mongolicus* [29]. Under the constraints of water and nutrients, increasing the input of N might be an adaptive strategy of plants, because increasing N could increase the number of leaf photosynthetic enzymes, thereby promoting photosynthesis [30]. The contents of P and K in each organ of *A. mongolicus* were generally low. On one hand, the strong wind erosion in the natural distribution area of *A. mongolicus* caused a large amount of clay and silt materials in the soil to be blown away, resulting in a large amount of loss of organic matter and nutrients and insufficient nutrients that could be absorbed and used by plant roots. On the other hand, K was an important nutrient element for plants to resist stress in arid climates, which could regulate plant water transmission and resistance to maintain basic metabolic activities [31]. The trade-off between the chemometric characteristics of plant organs reflected the regulatory strategies for plants to acquire resources and allocate nutrients in different habitats.

The characteristics of ecological stoichiometric ratios had an indicative effect on plant growth and nutrient supply. The larger the ratio of C:N and C:P in leaves, the stronger the ability of plants to assimilate C, reflecting the absorption and utilization efficiency of nutrient elements by plants. A large number of studies have shown that the stoichiometric characteristics of elements had a strong correlation with the growth rate of plants [32]. Rapidly growing organs have low C:P and N:P, whereas P content tends to increase [33]. In this study, the ratios of C:N, C:P, C:K and N:P in flowers and seeds were significantly lower than those in other organs, whereas the contents of N, P and K were significantly higher than those in other organs. In order to ensure the faster metabolism rate of the flower and the seed, the plant allocates more of its nutrients to the reproductive organs. The content of the elements and their stoichiometric ratio characteristics were in line with the growth rate hypothesis to a certain extent. The composition and stoichiometric ratio of C, N, P and K in plants were interrelated. Additionally, the relationship between this connection and the external environment jointly determined the growth and development of plants and their nutrient levels.

### 3.2. Response of Stoichiometric Characteristics of A. mongolicus Organs to Habitat

Habitat change affected physiological and biochemical processes such as the absorption, transportation, distribution and storage of C, N, P and K in plants [34,35,36]. In this study, there were significant differences in the contents of chemical elements among various organs of *A. mongolicus*. Leaves usually had higher nutrient content than stems and roots to maintain the high physiological and ecological activity of leaves [37,38]. As the transport channel, stems were mainly responsible for transporting nutrients from the root to the leaf, so the stored nutrients were the least. The root was the organ that absorbed and transported nutrients. The nutrient stored by roots was mainly used to support the growth and development of plants [39], so the C, N, P and K content in the order of leaf > root > stem. The variation of C content in the same organ of *A. mongolicus* in different habitats was relatively small, which was mainly due to the high content of C in each organ. As a structural material of plants, C mainly played a skeletal role in plants and did not directly participate in various production activities such as plant growth and reproduction, so it had good stability in plants. However, the N, P and K contents in each organ of *A. mongolicus* were significantly different in different habitats. N, P, and K were mainly enriched in reproductive organs, which provided sufficient material guarantee for the maturation of seeds [40]. The transition from vegetative growth to reproductive growth is to ensure the growth and reproduction of its population. The differential distribution of nutrients in vegetative growth and reproductive growth was closely related to the internal structure of plant organs and the differentiation of tissue functions, and it also reflects resource allocation patterns and plant growth strategies [1,41], and habitat changes might potentially regulate and influence this distribution process.

The heterogeneity of habitats led to the heterogeneity of physical and chemical properties in the soil, which led to the changes in the chemometric characteristics of plant organs [42,43]. Fixed and semi-fixed sandy land were sandy soil, which could reduce water evaporation, improve soil water holding capacity, facilitate the transformation and circulation of soil nutrients, and be effectively absorbed by plants. *A. mongolicus* had high soil nutrients and water, which was greatly beneficial to population renewal and plant growth. The plants with rich resources in the environment invest more resources in the aboveground parts. *A. mongolicus* shrubs were tall and had a large seed yield. The high nutrient tends to produce high-density vegetation, so the community coverage was large. Stony sandy land and alluvial gravel slope were gravelly soil with fast evaporation and poor soil water retention, which was not conducive to the maintenance of soil moisture and nutrients. Therefore, soil nutrients and moisture in this type of habitat were relatively low (Table 2), and plant growth slowed down. The height and crown width of *A. mongolicus* in gravelly desert were significantly lower than those in sandy desert. The community coverage was small. C:N and C:P can be understood as N- and P-nutrient-utilization efficiency. Generally, the higher the ratio is, the higher the utilization efficiency of N and P is [44]. The results showed that C:N, C:P and N:P of *A. mongolicus* in gravelly desert were significantly higher than those in sandy desert, indicating that *A. mongolicus* had higher unit nutrient productivity in stony–sandy land and alluvial gravel slope. This might be because *A. mongolicus* belongs to heliophilous species, which was drought and barren tolerant, and had higher requirements for climatic conditions than soil nutrients [15]. Although the nutrient content of fixed and semi-fixed sandy land were not the highest (Table 2), but their air permeability and water permeability were better than those of saline–alkali land. The unblocking flow of air and nutrients directly affected the resistance of plant roots when they penetrated the soil pores. As a deep-rooted species, *A. mongolicus* had a strong penetration ability of main roots. Its upward nutrient supply mainly depended on the absorption and supply of lateral roots distributed on the surface soil [18]. The drainage of saline–alkali land, which was not as strong as the permeability of fixed and semi-fixed sandy land, was poor and was not conducive to root growth. Therefore, *A. mongolicus* growth was worse in saline–alkali land, and its N and P utilization efficiency was the lowest, thus showing more serious restrictions. The C:K and N:K of *A. mongolicus* in the saline–alkali land with relatively good water conditions were significantly higher than those in other habitat types. This phenomenon was mainly due to the fact that under other habitat conditions with stronger water stress, *A. mongolicus* needed to control the transpiration of plants by increasing the K content, so as to enhance the drought resistance of plants.

## 4. Materials and Methods

### 4.1. Sample Area Description

Ulan Buhe Desert (39°16′48″–40°46′12″ N, 105°24′0″–107°1′12″ E) is located in the territory of Alxa League and Bayannaoer City, Inner Mongolia Autonomous Region. It is bordered by Langshan Mountain in the northwest, Hetao Oasis in the northeast, and the Yellow River in the east. The west is bounded by the Jilantai–Tukumu Highway in Alxa Left Banner, and the south is near the northern foot of Helan Mountain. The terrain is low in the middle and high at the edges, the southern area is higher than the northern area, and the interior is flat (Figure 3). The study area belongs to a typical mid-temperate continental arid monsoon climate zone, with little rainfall, intense evaporation, high temperature and high heat. The area has an average annual rainfall of 141 mm and an average annual evaporation of 2372 mm, which is more than 16 times the rainfall. On average, precipitation is mostly concentrated in June–September, with the characteristics of rain and heat in the same period. The annual average temperature is 7.8 °C, the annual sunshine time is 3229.9 h, and the temperature difference between day and night is large. It is one of the regions with the most sunshine hours in the country. The average wind speed is about 3.7 m/s [45], the average number of strong wind days for many years is more than 30 days, and the sandstorm season is between November and May of the following year. The landform types mainly include deserts, Gobi and mountains. The landforms are mainly fixed dune, semi-fixed dune and mobile dune, followed by alluvial gravel slopes distributed in front of mountains. Soil types included mainly aeolian sandy soil, grey desert soil, irrigation silt soil, saline soil and light brown calcium soil. As a typical oasis–desert transition zone in arid regions, the vegetation types are mainly shrubs and semi-shrubs with strong stress resistance. The shrubs are accompanied by typical desert steppe sandy, xerophytic annual and perennial herbs and artificial arbor forests that are drought-resistant and saline–alkali-tolerant.

### 4.2. Plot Selection and Setting

*A. mongolicus* is one of the main natural vegetation types in the Ulan Buhe Desert, widely distributed on the soil types with obvious zonality from Langshan Mountain to the northeastern edge of the Ulan Buhe Desert. This area belongs to the typical oasis–desert transition zone in arid regions. In the structural model of the Yinshan-piedmont gravel Gobi belt-desert-oasis, five habitat types are selected in the belt: fixed sandy land, semi-fixed sandy land, stony–sandy land, proluvial gravel slope land and saline–alkali land (Figure 4). Five typical sample plots of *A. mongolicus* community were set up in each plot, and 20 standard shrubs of *A. mongolicus* with good growth and no pests and diseases were selected in each sample plot, and their plant height, ground diameter, crown width and other index parameters were measured, and the community was recorded at the same time. The main associated species and vegetation coverage and other information (Table 2).

### 4.3. Sample Collection and Determination Methods

Flowers were collected in early May 2021, and organ samples including leaves, stems, roots and seeds were collected in mid-June. In each of the five habitat types, 10 *A. mongolicus* shrubs with good growth were selected, and sampling was carried out according to the different organs. The root system was collected by the tracking method. First, find the taproot of *A. mongolicus*, then dig down gradually along the direction of the taproot, cut off enough complete root system with branch shears, and place the root system sample of each plant of *A. mongolicus* in a ziplock bag for marking. Take a well-grown branch from each of the four directions of the stem. Select leaves with the top of the branches when they are fully extended, mature and free of pests and diseases. Cut them with scissors and place them in paper bags. In the present study, flowers and seeds were randomly collected from more than 30 shrubs and placed in mesh bags, and all the collected samples were brought back to the laboratory. Fix the plant samples at 85 °C for 30 min, and dry them at 65 °C for 48 h. Take the dried plant organ samples (>1 g) for coarse crushing with a cup mill. Then, finely crush them with a refrigerated mixing ball mill, pass the crushed samples through a 100-mesh sieve and bag and label them for experimental analysis. The carbon content was determined by the potassium dichromate external heating method. After being digested by the H_2_SO_4_-H_2_O_2_ method, the total nitrogen content was determined by the Kjeldahl method, the total phosphorus content was determined by the molybdenum–antimony colorimetric method, and the total potassium content was determined by the flame photometer method [46].

### 4.4. Statistical Analysis

We measured the element content of each organ (leaf, stem, root, flower and seed) and its stoichiometric ratio. We then calculated the following for the whole shrub: C, N, P, K, C:N, C:P, C:K, N:P, N:K and K:P, weighted by the proportional mass of each organ. The calculation was performed as follows: weighted mean value of shrub = weight × proportion of leaf + weight × proportion of stem + weight × proportion of root + weight × proportion of flower + weight × proportion of seed. We conducted descriptive analysis on the element content and stoichiometric ratio of each organ and the whole shrub of *A. mongolicus*. We used one-way analysis of variance (ANOVA) to test whether the stoichiometric characteristics of *A. mongolicus* were significantly different between organs and habitats. We used general linear models (GLM) to determine the effects of organs and habitats and their interaction on the stoichiometric characteristics of *A. mongolicus*. Data processing, graphing and statistical analysis were performed using Excel 2016, SPSS 18.0 and R software.

## 5. Conclusions

There are complex interactions between the element contents and stoichiometric ratios of each organ in *A. mongolicus* in the heterogeneous desert habitats. Plants allocate more nutrients to vigorously growing organs. The distribution of N, P and K elements is as follows: seed > flower > leaf > root > stem. The enrichment of nutrient elements in reproductive organs can promote the transition from vegetative growth to reproductive growth and provide sufficient material for the maturation of seeds. *A. mongolicus* mainly relies on seeds for reproduction but can coordinate a vegetative-growth and reproductive-growth relationship to maintain the growth and reproduction of its population. Compared with other studies, the content of C and N in the *A. mongolicus* in the present study is higher, whereas the content of P and K is relatively low, and the element content in the aboveground part is higher than that in the underground part. The N:P ratio of leaves in different habitats was greater than 16, indicating that the growth was mainly limited by P. The correlation analysis of element content in different organs showed that the nutrient synergy between different organs was much higher than that in the same organs. The contents of C, N, P and K and their stoichiometric ratios in different organs vary in different habitat types. The storage capacity of C, N and P is higher in sandy soil (fixed and semi-fixed sandy land), leading to taller plants and higher seed production, whereas the content of K is higher in gravelly soil (stony–sandy land and alluvial gravel slope), and the C:N, C:P and N:P ratios are significantly higher in gravelly soil than those in sandy soil. *A. mongolicus* has higher nutrient use efficiency in stony–sandy land and alluvial gravel slope. However, the plant growth is slower in the gravelly desert, so the plant height and crown width are lower, and the community coverage is smaller than in the sandy desert. In saline–alkali land, *A. mongolicus* has the lowest N and P utilization efficiency and poor growth. This is an important trade-off mechanism for plants to maintain growth and use nutrients efficiently.

## Figures and Tables

**Figure 1 plants-12-00414-f001:**
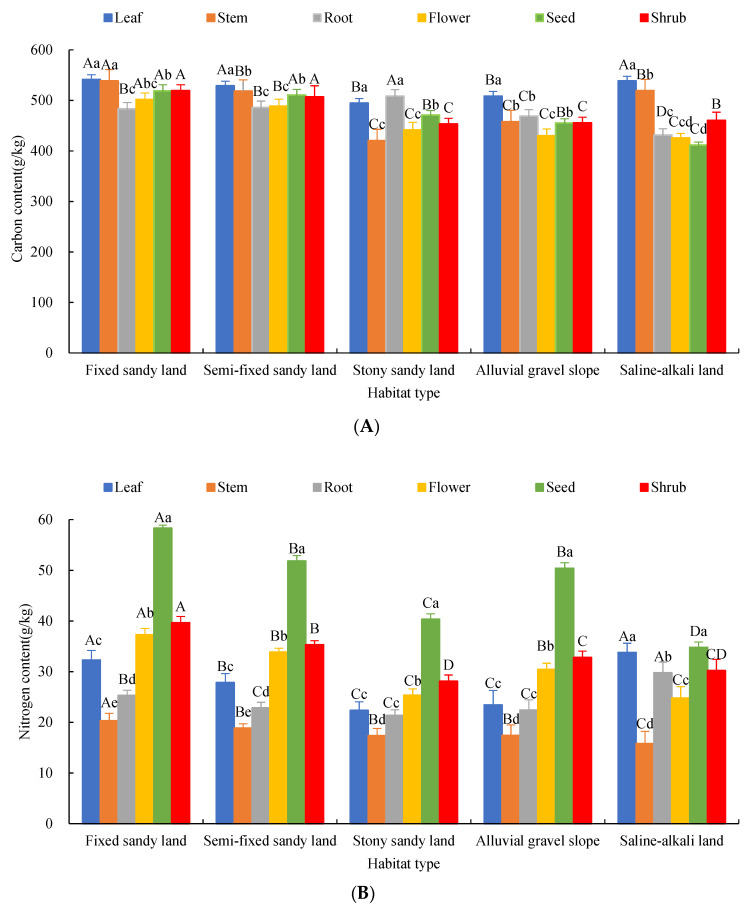
Carbon (**A**), Nitrogen (**B**), Phosphorus (**C**) and Potassium (**D**) concentrations in the organs of *A. mongolicus* in different habitats. Note: Upper case letters represent significant differences in different habitats of the same organ (*p* < 0.05). Lower case letters represent the significant characteristics of different organs in the same habitat (*p* < 0.05).

**Figure 2 plants-12-00414-f002:**
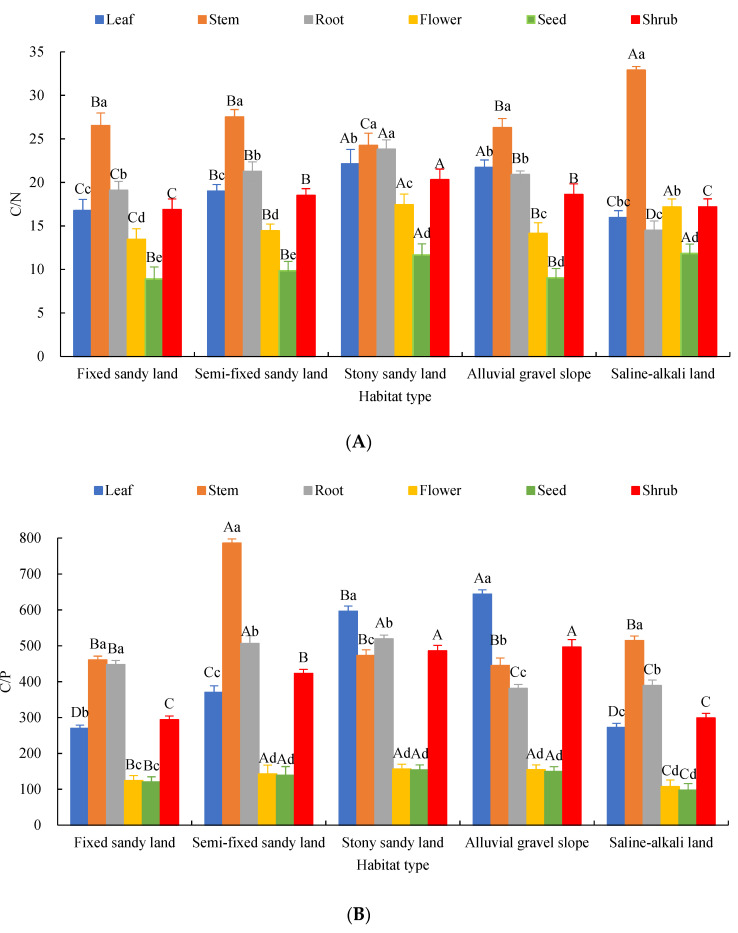
Elemental stoichiometric ratios in the organs of *A. mongolicus* in different habitats. (**A**) C/N, (**B**) C/P, (**C**) C/K, (**D**) N/P, (**E**) N/K and (**F**) K/P were all element stoichiometric ratios. Note: Upper case letters represent significant differences in different habitats of the same organ (*p* < 0.05). Lower case letters represent the significant characteristics of different organs in the same habitat (*p* < 0.05).

**Figure 3 plants-12-00414-f003:**
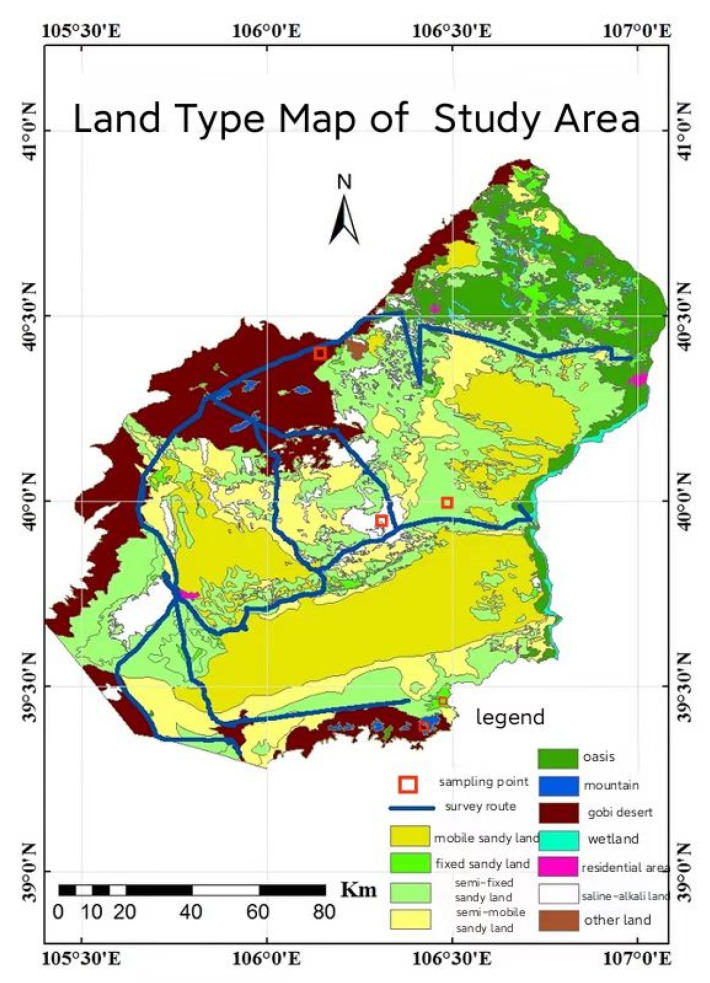
Study area of *A. mongolicus* in different habitats.

**Figure 4 plants-12-00414-f004:**
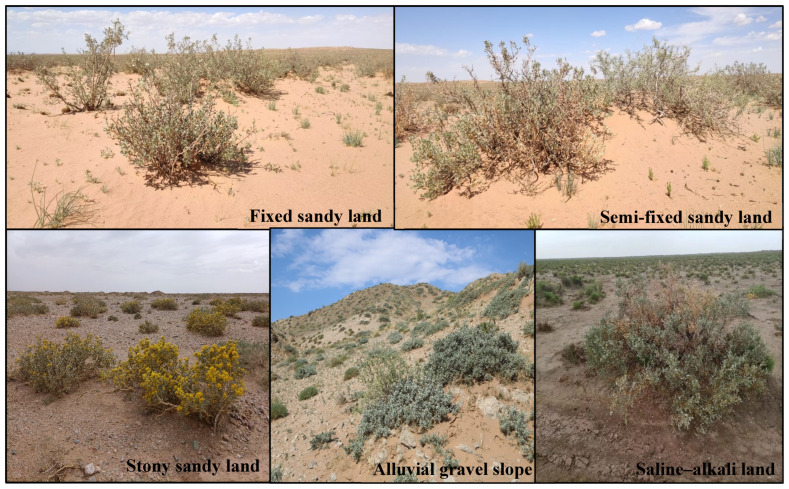
Five habitat types of *A. mongolicus.*

**Table 1 plants-12-00414-t001:** GLM analysis of habitat and organ effects on C, N, P and K contents and their stoichiometric characteristics.

Variables	Organ	Habitat	Interaction
C content	3.68 *	4.88 *	10.27 **
N content	79.81 **	15.26 **	8.91 **
P content	113.01 **	9.98 **	2.18
K content	399.00 **	147.24 **	4.87 **
C/N	157.81 **	3.45	14.49
C/P	42.14 **	2.59	3.59 *
C/K	101.28 **	55.15 **	11.95 **
N/P	37.39 **	3.60	5.24 **
N/K	42.11 **	52.29 **	18.63 **
K/P	17.51 **	116.37 **	13.43 **

Note: * indicates significant correlation (*p* < 0.05); ** indicates highly significant correlation (*p* < 0.01).

**Table 2 plants-12-00414-t002:** Plot basic characteristics of *A. mongolicus* in different habitats.

Habitat	Height (cm)	Diameter (mm)	The Crown Area (m^2^)	Vegetation Coverage (%)	Soil Moisture (%)	Soil Organic Carbon (g/kg)	Soil Total Nitrogen (g/kg)	Soil Total Phosphorus (g/kg)
Fixed sandy land	118.23 ± 19.65	24.39 ± 2.89	12.51 ± 1.89	25–40%	3.92 ± 0.75	3.92 ± 1.75	0.38 ± 0.17	0.34 ± 0.06
Semi-fixed sandy land	89.5 ± 3.7	22.69 ± 2.33	8.23 ± 0.45	10–25%	3.36 ± 1.02	3.36 ± 1.07	0.31 ± 0.08	0.32 ± 0.05
Stony sandy land	67.6 ± 5.6	15.72 ± 1.05	5.68 ± 0.56	15–30%	3.25 ± 0.83	3.35 ± 0.93	0.26 ± 0.04	0.27 ± 0.03
Alluvial gravel slope	34.5 ± 2.9	7.89 ± 1.03	2.09 ± 0.42	10–25%	2.91 ± 0.72	2.99 ± 0.82	0.29 ± 0.07	0.31 ± 0.02
Saline–alkali land	78.6 ± 6.6	18.75 ± 1.19	6.48 ± 0.56	20–35%	4.39 ± 0.85	4.19 ± 0.45	0.41 ± 0.12	0.36 ± 0.04

Note: The area of crown = π × crown major axis × crown minor axis. The data in the table are mean ± standard deviation.

## Data Availability

All data are presented in the main text.

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
