# Peer review of "Ecological Stoichiometric Characteristics in Organs of Ammopiptanthus mongolicus in Different Habitats"

_plants, 2023, doi:10.3390/plants12020414_

Round 1
Reviewer 1 Report
The topic is interesting but the paper needs major revisions before acceptance for publication. The following issues need to be addressed to improve transparency and communication of the results:
1. Line 17 &75 my country? Which country? Please clarity!
2. Generally, the research on stoichiometric just focused on C, N, P, while you mentioned K, please explain why?
3. Line 58 delete the ‘.’ between “plant growth” and “and nutrient”
4. Some research results and conclusions on stoichiometry should be further introduced in the Introduction.
5. Does the Ammopiptanthus mongolicus exist only in China? Please clarity!
6. The aim of this research should be mentioned at the end of the Introduction
7. A map for the location of the study site and the sampling sites should be added for better understanding the information of study region.
8. Line 163 please delete the “. content”.
9. Line 170 please delete the “:” and upper case “The”.
10. What is the mean of “Aa” in figure 2? I think the Upper and lower case letters describes different meanings?
11. Too much information in Figure 2, I suggest dividing it into two figures. One for concentrations and the other for proportion.
12. More than one graph contained in Figure 2 &3, so please give a code for each subgraph.
13. Discuss for Discussion
14. What is your results suggest for? Please added in the Discussion.
Author Response
Please see the attachment.
Thank you very much for your valuable suggestions and comments.
Kind regards,
Dong xue

Reviewer 2 Report
Dear authors,
I have reviewed your manuscript “Ecological stoichiometric characteristics in organs of Ammopiptanthus mongolicus in different habitats”
MS falls within the journal’s scope and the subject matter is quite interesting. However, the Abstract is too lengthy and needs to rewrite precisely. The Introduction is failed to motivate and problematize the objectives of study, so it should be improved. The methodology needs to be revised and the statistical methods/tools/software used for data analysis need to be discussed in detail. In addition, the results are not presented in a good way. Moreover, extensive editing of the English language, grammar and style is required, and there are several drawbacks to address. Briefly, the article can't accept in its present form.
Some comments and suggestions about MS
Title
Line 2-3: The title of manuscript is good.
Abstract
Line 10-49: According to journal (Plants) policy, the abstract should not be more than 200 words and your abstract is more than 600. So please rewrite the precise and short Abstract to meet the journal’s requirements and for easy understanding of readers.
Line 48-49: Please avoid repeating the same words (like Ammopiptanthus mongolicus; ecological stoichiometry; organ) that you have already used in the title to enhance the visibility of your article.
Introduction
The authors are encouraged to cite recent literature in the introduction.
https://doi.org/10.1016/j.jplph.2022.153671 ,
https://doi.org/10.3390/plants9080990.
Line 52-58: This is a lengthy and ambiguous sentence, so please rephrase and divide the sentence into small sentences for easy understanding.
Line 63-63: I don't see the use of this sentence. So, please delete it.
Materials and Methods
2.1. Sample Area Description
Please add the study area map.
Line 116: Provide the reference for this statement ‘The average wind speed is about 3.7 m/s’
2.4. Statistical Analysis
Line 164-175: Punctuation and grammatical issues need to address.
Results
Results presentation is not good.
Line 182: "P" should be in small letter and italicized. Please check and correct this throughout the manuscript.
Line 214-215: Figure 1 &2 are mixed up, where is the legend of Figure 1?
Line 246: You made 6 subfigures for Figure 3, properly assign them numbers/letters. Same comments for other figures.
Discussion
Discussion is shallow.
Conclusions
Conclusions need to be revised as per factual findings should be to the point, comprehensive and logical.
In light of the above, I think the MS deserves to be published but should only be accepted after a Major Revision.
Please take into consideration the comments on the PDF file of the revision.

Author Response

(The authors gave the same response as above.)

Round 2
Reviewer 1 Report
The author made a good revision, I think the Materials and Methods should be the second part.
Author Response
My article belongs to the special issue "Sand Vegetation and Restoration". Its format requires that Materials and Methods be placed in fourth part .
Thank you for your comments and suggestions during the revision of my article. At the same time, it is also very helpful for my next research.
Kind regards,
Dong xue
Reviewer 2 Report
Thanks to the authors for critically addressing my concerns. I am satisfied with the current version. However, the references should be formatted according to the journal style before possible acceptance/publication in the ‘PLANTS’.
Author Response
I will strictly follow the format of the journal style references.
Thank you for your comments and suggestions during the revision of my article. At the same time, it is also very helpful for my next research.
Kind regards,
Dong xue